# Dual genome-wide CRISPR knockout and CRISPR activation screens identify mechanisms that regulate the resistance to multiple ATR inhibitors

**Emily M. Schleicher**, **Ashna Dhoonmoon**, **Lindsey M. Jackson**, **Kristen E. Clements**, **Coryn L. Stump**, **Claudia M. Nicolae**, **George-Lucian Moldovan***

Department of Biochemistry and Molecular Biology, The Pennsylvania State University College of Medicine, Hershey, Pennsylvania, United States of America

* glm29@psu.edu

**Data Availability Statement:** All relevant data are within the manuscript and its Supporting Information files.

## Abstract

The ataxia telangiectasia and Rad3-related (ATR) protein kinase is a key regulator of the cellular response to DNA damage. Due to increased amount of replication stress, cancer cells heavily rely on ATR to complete DNA replication and cell cycle progression. Thus, ATR inhibition is an emerging target in cancer therapy, with multiple ATR inhibitors currently undergoing clinical trials. Here, we describe dual genome-wide CRISPR knockout and CRISPR activation screens employed to comprehensively identify genes that regulate the cellular resistance to ATR inhibitors. Specifically, we investigated two different ATR inhibitors, namely VE822 and AZD6738, in both HeLa and MCF10A cells. We identified and validated multiple genes that alter the resistance to ATR inhibitors. Importantly, we show that the mechanisms of resistance employed by these genes are varied, and include restoring DNA replication fork progression, and prevention of ATR inhibitor-induced apoptosis. In particular, we describe a role for MED12-mediated inhibition of the TGFβ signaling pathway in regulating replication fork stability and cellular survival upon ATR inhibition. Our dual genome-wide screen findings pave the way for personalized medicine by identifying potential biomarkers for ATR inhibitor resistance.

## Author summary

Cancer cells rely on the ATR replication stress response pathway to ensure DNA replication and continued cellular proliferation. As such, inhibitors of the ATR kinase activity represent promising new anti-cancer drugs. However, the tumors' susceptibility to these drugs can be markedly impacted by their genomic profile. To address this, we employed dual CRISPR knockout and activation genome-wide genetic screens to catalog the genetic determinants of the cellular resistance to multiple ATR inhibitors. We identified several mechanisms which control this resistance, including regulation of apoptosis and stabilization of replication fork stability. Our work lays the foundation for personalized deployment of ATR inhibitors in cancer therapy.

**Funding:** This work was supported by: NIH R01ES026184 and R01GM134681 (to GLM) and 1F31CA243301 (to EMS). The funders had no role in study design, data collection and analysis, decision to publish, or preparation of the manuscript.

**Competing interests:** The authors have declared that no competing interests exist.

## Introduction

Proper response to DNA damage and replication stress is critical for all organisms. Replication stress occurs upon arrest of the DNA replication machinery at sites of DNA damage, or during replication of endogenous difficult to replicate DNA sequences such as microsatellite regions [1]. The ataxia-telangiectasia-mutated (ATM) and ataxia telangiectasia and Rad3-related (ATR) kinases are in primary control of the cellular responses to replication stress and DNA damage [2]. The activation of these kinases is critical to arrest the cell cycle and allow time for proper execution of DNA replication and repair prior to cell division [3]. ATR is activated by single-stranded DNA (ssDNA) formed upon replication fork arrest. ATR activation leads to downstream phosphorylation of Chk1, resulting in stabilization of the replication fork, suppression of origin firing, and cell cycle arrest. ATM is triggered by the presence of double-stranded DNA breaks, and phosphorylates p53 and Chk2 leading to cell cycle arrest to allow for the DNA to be repaired before proceeding through the cell cycle.

Cancer cells heavily rely on the replication stress response for viable cell division [4]. Many of the current cancer treatment agents are genotoxic compounds that lead to a variety of adverse side effects for patients, as non-tumor cells in the body can also be affected. One way to avoid these side effects is to enhance the specific targeting of cancer cells, by exploiting their reliance on the replication stress response. Thus, targeting ATR has been proposed as a potential cancer therapy [5]. ATR inhibitors (ATRi) could be an option for killing cancer cells with an inherently large amount of DNA damage, due to the pivotal role of ATR in the DNA damage response [6]. Additionally, non-tumor cells in the body have little to no replication stress, and thus should not be affected by ATRi. ATR inhibitors may also be used in combination with DNA damaging agents as a therapeutic option [7, 8]. Moreover, since ATR and ATM work together to manage DNA damage within the cell, ATR inhibitors have been shown to be particularly efficient in ATM/p53 deficient tumor cells [9–11].

Since the development of ATR inhibitors, the effects of ATR/Chk1 pathway inhibition have been examined more closely. It was originally shown that inhibition of Chk1 caused a decrease in the inter-origin distance during DNA replication, accompanied by a decrease in replication fork progression [12]. Under normal conditions, Chk1 inhibits replication initiation by blocking CDC45 recruitment to the MCM2-7 complex, which is necessary for unwinding DNA at the replication fork [13]. Much like loss of Chk1, ATR inhibition has been shown to cause a decrease in replication fork speed as well as an increase in the amount of origins that are firing during DNA replication [14, 15].

AZD6738 and VE822 are two orally bioavailable, ATP competitive ATR inhibitors, which block the phosphorylation of Chk1, a downstream target of ATR, but do not significantly inhibit ATM [11, 16, 17]. AZD6738 and VE822 are currently in 11 phase I and phase II clinical trials and are being tested alone or in combination with other drugs such as Olaparib, Cisplatin, Paclitaxel, etc [18, 19]. In addition, some of the current clinical trials focus on patients with particular tumor biomarkers such as DNA damage response pathway mutations, DNA damage, p53 mutations, or ATM deficiency [19]. Early results of these clinical trials show that both ATR inhibitors have low toxicity in patients and work best in combination with other drugs [20, 21]. Specifically, VE822 was shown to be well tolerated in phase I trials with low toxicity and no dose-limiting affects [22]. Similarly, AZD6738 was shown to have reduced toxicity, both alone and in combination with a PARP inhibitor [22].

Much like with other cancer therapies, identifying the subsets of tumors that respond well or are resistant to the drug will become paramount for efficient use of ATRi in the clinic. Genome-wide screens are an effective technique to identify biomarkers of drug resistance and sensitivity. Recently reported genome-wide CRISPR knockout screens identified genetic

determinants of ATRi sensitivity [23–25]. By using a relatively low ATRi dose, these screens were designed to identify genes that, when lost, caused sensitivity to AZD6738. In contrast, little is known about genes that cause resistance to ATR inhibitors when inactivated. Moreover, these previous screens only investigated one ATR inhibitor. Additionally, genome-wide identification of genes that alter the response to ATRi when overexpressed rather than suppressed, has not yet been addressed.

To comprehensively identify the genes regulating the resistance to ATR inhibitors, we employed a dual CRISPR screening approach wherein we investigated both loss and overexpression of the majority of genes in the human genome. We performed both knockout and activation screens in HeLa cancer cells. Additionally, we performed the activation screens in non-transformed, breast epithelial MCF10A cells. All screens were performed separately with two different ATR inhibitors, VE822 and AZD6738. This comprehensive approach allowed unbiased identification of genes that affect ATRi resistance.

## Results

### Genome-wide CRISPR knockout screens identify genes regulating the resistance to multiple ATRi

A dual genome-wide CRISPR knockout and activation screening approach was designed to identify genes involved in the response to multiple ATR inhibitors (Fig 1). First, to identify genes whose loss confers resistance to ATRi, the Brunello human CRISPR knockout lentiviral library was employed [26]. This library targets 19,114 genes with 76,441 unique guide RNAs (gRNAs), thus on average covering each gene with four different gRNAs. To maintain 250-fold library coverage, 20 million library-infected cells were treated with VE822 (1.5μM), AZD6738 (3.6μM), or DMSO control. In contrast to previously published screens which focused on ATRi sensitivity and thus used a relatively low dose, we chose these high ATRi dosages as we previously determined that they kill approximately 90% of cells over 108 hours of treatment, thus allowing us to specifically study resistance to the drugs. Surviving cells were collected and genomic DNA was extracted. The gRNA sequences were PCR-amplified and identified by Illumina sequencing (Fig 1A and 1C). Bioinformatic analyses using the redundant siRNA activity (RSA) algorithm [27] was used to generate separate ranking lists of genes that were enriched in the VE822 and AZD6738 conditions compared to the control (S1 Table). This represents genes that, when inactivated, confer resistance to ATRi. Interestingly, there was large overlap between the VE822 and AZD6738 screen results, indicating common response mechanisms to the two different ATRi. Of the top hits of each of the two ATRi screens with logP values lower than -2.0 (393 genes for the VE822 screen and 456 genes for the AZD6738 screen), 118 were present in both of them (Fig 2A and S2 Table), which is much higher than the random probability (Fig 2B). Moreover, 7 genes were common within the top 40 hits of each ATRi screen. Biological pathway analysis of the top hits of both screens with logP values lower than -2.0 revealed common biological processes, including protein translation, DNA replication, and sister chromatid cohesion (Fig 2C). Notably, multiple components of the cell cycle, cell migration, and DNA repair biological processes were found in both ATRi screens (Fig 2D).

### Validation of hits from the knockout screen

Seven common genes were found within the top 40 genes in both ATRi resistance CRISPR knockout screens; these genes were: KNTC1, EEF1B2, LUC7L3, SOD2, MED12, RETSAT, and LIAS (Fig 3A and 3B). None of these genes were previously shown to induce ATRi resistance, to our knowledge. Thus, we sought to directly confirm that these seven genes alter the ATRi

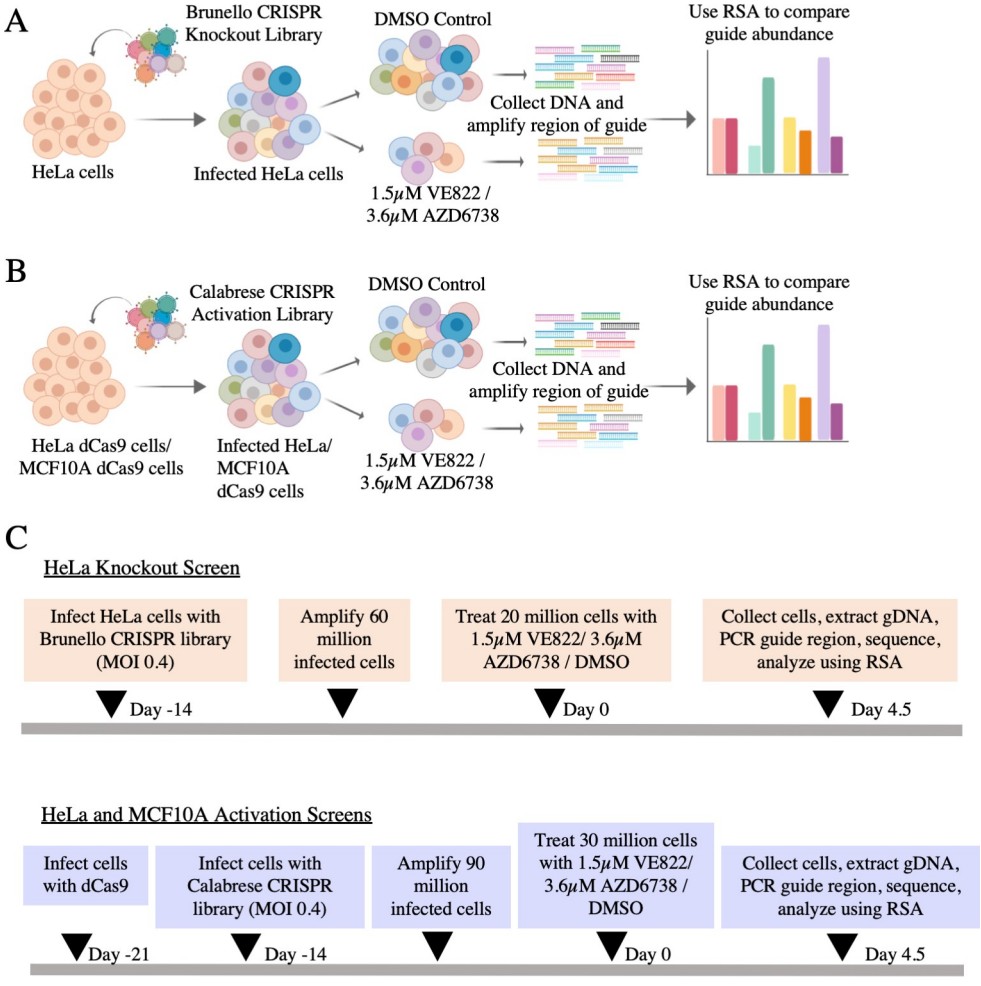

**Fig 1. Overview of the dual CRISPR knockout and CRISPR activation screens to identify genes that regulate the resistance to the ATR inhibitors AZD6738 and VE822.** (**A**) CRISPR knockout screens performed in HeLa cells using the Brunello CRISPR knockout library. (**B**) CRISPR activation screens performed in HeLa and MCF10A cells using the Calabrese CRISPR activation library. (**C**) Timelines of the knockout and activation CRSIPR screens.

response. First, we tested these genes in HeLa cells, in which the screen was originally performed. We employed siRNA to knockdown each of these genes. The knockdown efficiency was confirmed by western blot for all seven proteins (S1 Fig). We performed clonogenic assays by incubating siRNA-treated HeLa cells with 1μM AZD6738 or 0.5μM VE822. After three days the media was replaced and colonies were allowed to grow for two weeks. HeLa cells with knockdown of each of the seven hits presented more colonies compared to control, indicating that they are more resistant to ATR inhibitors (Fig 3C). Moreover, we also validated the seven top hits by measuring cellular proliferation in the presence of ATRi. HeLa cells were treated with siRNA and incubated for three days with 0.5μM VE822, 1μM VE822, 0.5μM AZD6738, or 1μM AZD6738. Cellular proliferation was determined using the CellTiterGlo reagent (Promega). Knockdown of each of the 7 top hits resulted in increased cell survival after ATR inhibitor treatment compared to control (Fig 3D). These findings show that all seven hits investigated confer resistance to both ATRi tested, thus validating our screen results.

We next sought to investigate if these findings are restricted to HeLa cells, or in fact these hits also regulate ATRi resistance in other cell lines. To address this, we repeated the cellular

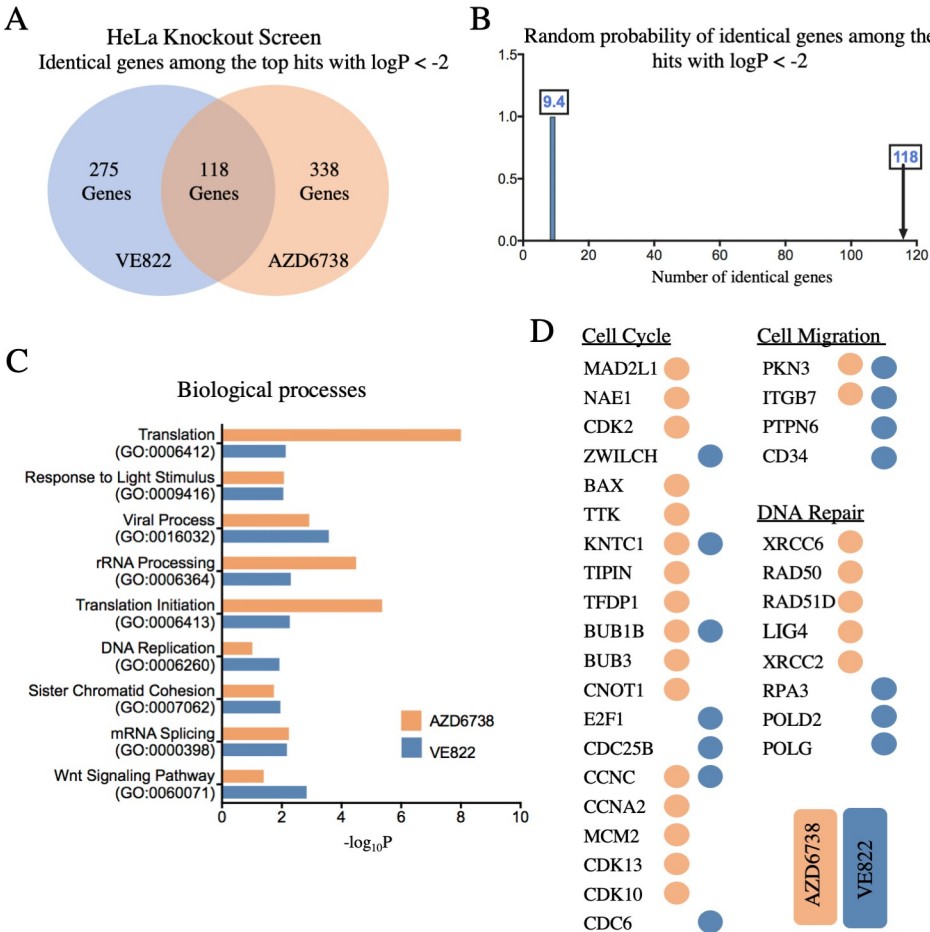

**Fig 2. Pathway analysis reveals biological pathways involved in resistance to AZD6738 and VE822 from the CRISPR knockout screens.** (**A**) Diagram showing the overlap of identical genes within the top hits from both screens with logP values lower than -2.0. (**B**) The number of common genes within the top hits with logP values lower than -2.0 (namely 118) is much higher than the random probability of identical hits, which is 9.4. (**C**) Biological pathways that were significantly enriched within the top hits with logP values lower than -2.0 from both ATRi screens using Gene Ontology analysis. (**D**) Genes within the top hits with logP values lower than -2.0 of each ATRi screen that are involved in the indicated biological processes.

proliferation experiments in MCF10A normal breast epithelial and 8988T pancreatic cancer cell lines. Both of these cell lines were slightly less sensitive to ATR inhibitors, so higher concentrations were used. Cells treated with 1μM VE822, 2μM VE822, 1μM AZD6738, and 2μM AZD6738 were analyzed for cellular proliferation after 3 days. In MCF10A cells, knockdown of all hits with the exception of SOD2 resulted in resistance to either ATR inhibitor (Fig 3E). Similar results were obtained for 8988T cells, in both cellular proliferation and clonogenic sensitivity assays (Fig 3F and 3G). These findings indicate that the genes identified by screening HeLa cells control ATRi resistance across multiple cell lines.

## Loss of the top hits does not restore ATR catalytic activity

One potential mechanism of ATRi resistance is the restoration of ATR catalytic activity in the presence of ATRi. Chk1 is the main component of the ATR signaling cascade that is activated by DNA damage [28]. Once ATR is activated, its kinase activity phosphorylates Chk1 leading

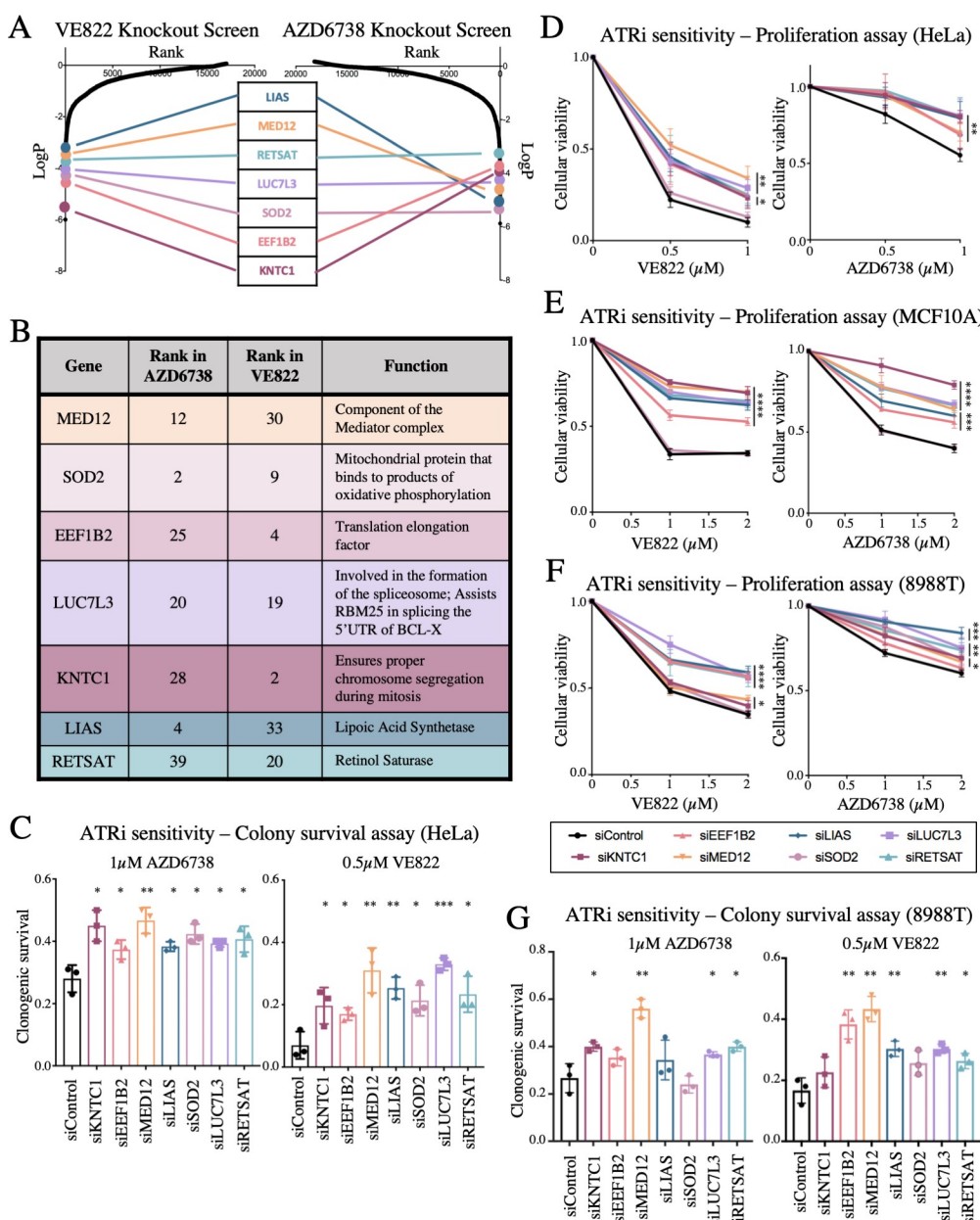

**Fig 3. The top seven common hits were confirmed to cause resistance to ATRi when depleted in three different cell lines.** (**A**) Scatterplot showing the results of the ATRi resistance knockout screens. Each gene targeted by the library was ranked according to P-values calculated using RSA analysis. The P-values are based on the fold change of the guides targeting each gene between the ATRi- and DMSO-treated conditions. There were seven identical hits in the top 40 hits of both ATRi knockout screens. (**B**) The seven top common hits have diverse biological functions. (**C**) Knockdown of the top common gene hits in HeLa cells results in resistance to ATRi in a colony survival assay. The average of three experiments is shown, with error bars representing standard deviations. Asterisks indicate statistical significance for each hit compared to control. (**D**) Knockdown of the top common gene hits in HeLa cells results in resistance to ATRi in a cellular proliferation assay. The average of three experiments is shown, with error bars representing standard deviations. Asterisks indicate statistical significance for each hit compared to control. (**E, F**) Knockdown of the top common gene hits also results in resistance to ATRi in MCF10A cells (**E**) and 8988T cells (**F**) in cellular proliferation assays. The average of three experiments is shown, with error bars representing standard deviations. Asterisks indicate statistical significance for each hit compared to the control. (**G**) Knockdown of the top common gene hits in 8988T cells results in resistance to ATRi in a colony survival assay. The average of three experiments is shown, with error bars representing standard deviations. Asterisks indicate statistical significance for each hit compared to control.

to all of the downstream effects. As a result, Chk1 phosphorylation at Serine 317 and Serine 345 represent markers of ATR cascade activation and activity. Thus, we knocked down the top hits and analyzed the levels of phosphorylated Chk1 by western blot, under normal (no treatment) conditions, hydroxyurea treatment, ATRi treatment, and a combination of hydroxyurea and ATRi. Hydroxyurea depletes the cellular dNTP pools, thereby stimulating the replication stress response and activating ATR. We found that knockdown of the top hits did not restore Chk1 phosphorylation at either Serine 317 or Serine 345 (S2A and S2B Fig). Similarly, there was no obvious difference in the level of γH2AX (S2C Fig). These results indicate that these genes do not act by restoring ATR activity in the presence of ATRi.

## Loss of LUC7L3 causes resistance to ATRi by suppressing ATRi-induced apoptosis through RBM25-mediated alternative splicing of BCL-X

Treatment of HeLa cells with VE822 for three days resulted in induction of apoptosis, as measured by detecting Annexin-V expression (Fig 4A). Thus, we sought to employ this assay as another readout of resistance to ATR inhibitors. Knockdown of each of the 7 top hits significantly reduced the amount of apoptosis induced by treatment with 0.5μM VE822 for three days (Fig 4B), further validating that their loss confers resistance to ATRi.

One of the top hits, namely LUC7L3, is a component of the U1 snRNP (small nuclear ribonucleoprotein) complex which affects 5′ splice site selection in yeast and human cells [29]. It was previously reported that LUC7L3 interacts with the splicing factor RBM25 (RNA-binding motif protein 25), which also scored highly in our screen (ranked #201 in the AZD6738 screen). Indeed, RBM25 knockdown in HeLa cells resulted in ATRi resistance, in both cellular proliferation and clonogenic sensitivity assays, similar to loss of LUC7L3 (Fig 4C and 4D). RBM25 is an RNA-interacting protein which binds to the exonic splicing enhancer sequence CGGGCA in the pre-mRNA of the apoptosis regulator BCL-X, to promote the alternative splicing from the anti-apoptotic BCL-X$_{long}$ isoform to the pro-apoptotic BCL-X$_{short}$ isoform [30–32]. While the impact of LUC7L3 on BCL-X splicing was not directly investigated, it was proposed that the RBM25-LUC7L3 interaction is important for recruitment of the U1 snRNP complex to the BCL-X pre-mRNA to initiate this alternative splicing [30]. Thus, we hypothesized that the LUC7L3-RBM25 complex may control ATRi resistance by regulating the levels of the BCL-X isoforms. To address this, we tested if LUC7L3 depletion affects BCL-X splicing in ATRi-treated cells. Loss of LUC7L3 or of RBM25 resulted in a reduction of the pro-apoptotic BCL-X$_{short}$ isoform, at both the mRNA and protein levels (Figs 4E and 4F and S3A and S3B). A corresponding increase in the anti-apoptotic BCL-X$_{long}$ isoform was detected (S3C Fig). Thus, our results validate the proposed role of LUC7L3 in RBM25-mediated BCL-X splicing, and indicate that this mechanism regulates the cellular resistance to ATR inhibitor treatment.

## Loss of MED12 and LIAS stabilizes the replication fork in response to ATRi

A major role of ATR is to stabilize replication forks upon genotoxic insults [33]. ATR inhibitors have been shown to cause a decrease in replication tract length [15]. This replication fork slowing is detrimental to the cell and may eventually result in replication deficiency and cell cycle arrest. We sought to investigate if any of the hits described above regulate ATRi resistance by suppressing the impact of ATRi on fork slowing. For this, we employed the DNA fiber assay. We incubated cells consecutively with two thymidine analogs to detect active DNA replication forks. IdU was added first for 30 minutes followed by CldU for 30 minutes. VE822 was added concomitantly with CldU. With increasing amounts of ATRi (2μM VE822, 4μM VE822, or 6μM VE822), the CldU replication tract was significantly decreased (Fig 5A), thereby confirming that ATR inhibition causes a decrease in replication fork progression.

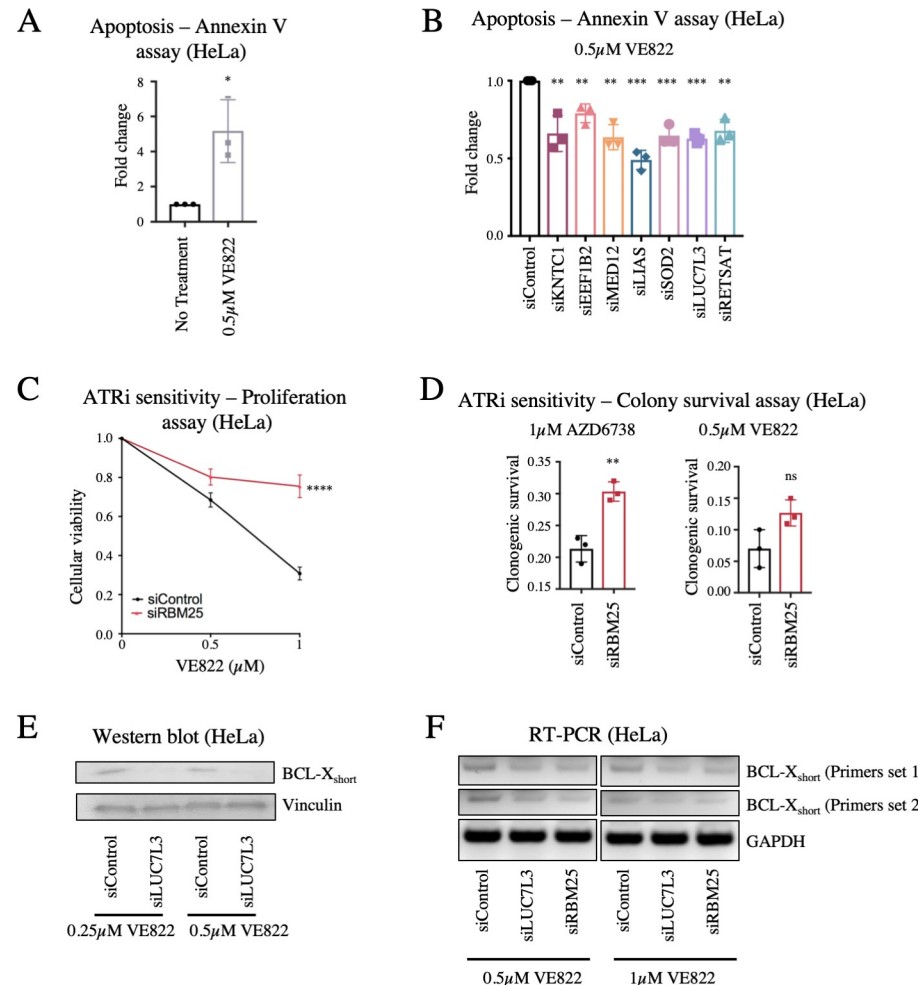

**Fig 4. Regulation of ATRi-induced apoptosis by the LUC7L3-RBM25 pathway.** (**A**) Treatment with ATR inhibitors results in apoptosis induction, as measured by the Annexin-V assay. The average of three experiments is shown, with error bars representing standard deviations. Asterisks indicate statistical significance. (**B**) Knockdown of the top 7 hit genes in HeLa cells causes a significant reduction in apoptosis after treatment with 0.5μM VE822 for 36 hours. First, data for each hit was normalized to their own untreated control, and then normalized to siControl. The average of three experiments is shown, with error bars representing standard deviations. Asterisks indicate statistical significance for each hit compared to control. (**C**, **D**) RBM25 knockdown in HeLa cells results in ATRi resistance in cellular proliferation (**C**) and colony survival (**D**) assays. The average of four (**C**) or three (**D**) experiments is shown, with error bars representing standard deviations. Asterisks indicate statistical significance. (**E**) Western blot showing a decrease in the protein levels of the BCL-X$_{short}$ isoform in HeLa cells after knockdown of LUC7L3 and ATRi treatment. (**F**) Reverse transcriptase PCR showing a decrease in BCL-X$_{short}$ mRNA after knockdown of LUC7L3 or RBM25 followed by ATRi treatment in HeLa cells. Two different primer sets for BCL-X$_{short}$ were used, and GAPDH was used as control.

Next, we investigated the impact of the top hits described above, by performing the DNA fiber combing assay with cells depleted of each of these genes by siRNA. Cells were treated with IdU for 30 minutes followed by co-treatment with CldU and 2μM VE822 for 30 minutes. The effect of loss of the top hits was investigated by calculating the ratio of CldU tract length to IdU tract length. Strikingly, knockdowns of MED12 and LIAS significantly increased the CldU-to-IdU ratio, indicating that fork slowing in the presence of ATRi is suppressed (Fig 5B and 5C). The other hits did not affect the CldU-to-IdU ratio. Therefore, the ATRi resistance observed upon loss of MED12 and LIAS may involve restoration of replication fork speed in

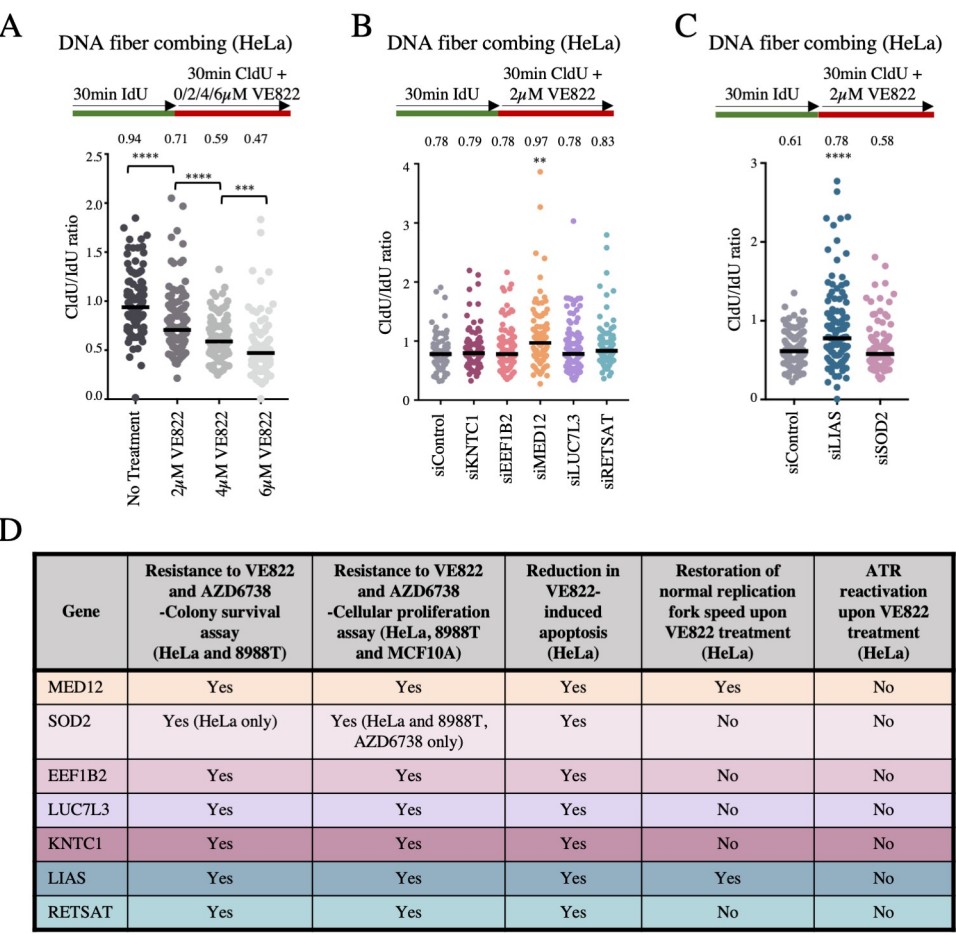

**Fig 5. Knockdown of MED12 and LIAS promotes replication fork stability in the presence of ATRi.** (**A**) The addition of increasing concentration of ATRi in HeLa cells causes a significant decrease in replication tract length, as measured by the ratio of the CldU tract length (with ATRi) to the IdU tract length (without ATRi). (**B, C**) Upon treatment with 2μM VE822 concomitant with CldU, knockdown of MED12 (**B**) or LIAS (**C**) showed a restoration of normal fork speed, as measured by the ratio of the CldU tract length (with ATRi) to the IdU tract length (without ATRi). The other hits do not affect fork slowing induced by ATRi. In all panels, the median values are indicated for each sample, and the asterisks indicate statistical significance. At least 100 fibers were quantified. (**D**) Table summarizing the impact on ATRi-induced cellular outcomes, observed for the seven top screen hits which were validated. "Yes" indicates differences which were statistically significant. The cell lines and ATRi used are indicated.

the presence of ATR inhibitors. In conclusion, we validated seven top hits involved in the cellular resistance to both ATR inhibitors, through likely distinct mechanisms which do not involve restoration of ATR activity (Fig 5D).

## MED12 modulates the cellular response to ATR inhibitors in a manner independent of the Mediator complex

MED12 is a member of the Mediator transcriptional co-activator complex, which is generally required for transcription by RNA polymerase II [34]. A recently published CRISPR screen designed to identify genes whose inactivation results in sensitivity to ATRi uncovered RNA-SEH2 as a top candidate [25]. RNASEH2 is one of the main factors involved in dissolution of R-loops, which are DNA-RNA hybrids formed during transcription [35, 36]. R-loops can potentially interfere with DNA replication progression [1, 37]. In yeast, the Mediator complex

was shown to regulate R-loop formation [38]. Since we observed that loss of MED12 restores replication fork speed upon ATRi treatment, one possibility we envisioned is that R-loops formed during transcriptional events dependent on the Mediator complex form replication blocks which are toxic to cells upon ATR inhibition unless they are resolved by RNASEH2. To investigate this possibility, we co-depleted RNASEH2 and MED12, and measured ATRi sensitivity. RNASEH2 depletion resulted in VE822 sensitivity not only in wildtype cells, but also in MED12-knockdown cells (Figs 6A; S4). Moreover, RNASEH2 depletion reduced replication fork speed in MED12-knockdown cells upon VE822 treatment (Fig 6B). These findings indicate that RNASEH2 activity protects against the damaging effects of ATRi not only in wildtype cells, but also in MED12-deficient cells, arguing against the model that MED12 loss promotes ATRi resistance by suppressing R-loop formation.

We next sought to investigate if Mediator complex-promoted transcription by RNA polymerase II promotes ATRi sensitivity. Inhibition of RNA polymerase II using α-amanitin did not impact ATRi resistance (S5 Fig), arguing against a general role for transcription in ATRi sensitivity. We next specifically investigated the potential involvement of the Mediator complex itself. The Mediator is composed of a large core complex, which is essential for its functions in promoting transcription, and an associated regulatory subcomplex known as the CDK8 module [34, 39]. Since MED12 is a member of the CDK8 module, we first investigated if the core complex is also involved in the cellular response to ATRi. Unlike MED12 downregulation, depletion of the core complex subunit MED7 did not affect the sensitivity to VE822 or AZD6738 (Figs 6C and 6D and S6A). We next investigated if the CDK8 module is specifically involved in ATRi sensitivity, as a separate function outside of the Mediator complex. In contrast to MED12 knockdown, but similar to MED7 kncokdown, depletion of the CDK8 module subunits MED13, CDK8 and CCNC (Cyclin C) did not markedly affect VE822 sensitivity (Figs 6E and 6F and S6B–S6D). Similar results were obtained for AZD6738 sensitivity (S6E Fig). Finally, unlike MED12 depletion, loss of core complex subunit MED7 or of the CDK8 module subunits MED13 and CDK8 did not restore normal replication fork speed upon VE822 treatment (Fig 6G). Overall, these findings indicate that MED12 exerts this novel role in the cellular response to ATRi independently of the Mediator core complex or of the CDK8 regulatory module that it is part of.

## Activation of the TGFβ pathway underlies the replication fork stability and cellular resistance to ATRi observed upon MED12 depletion

MED12 was previously identified as a top candidate in an RNAi screen investigating the genetic determinants of the cellular response to inhibition of ALK and EGF receptor tyrosine kinases [40]. Similar to the situation we describe here for ATRi, it was shown that depletion of MED12, but not of other components of the core Mediator complex or of the CDK8 module, results in resistance to ALK and EGF receptor inhibitors. This resistance was found to depend on activation of the TGFβ pathway, and MED12 was shown to inhibit the glycosylation of the TGFβ receptor TGFBR2 and thereby block its expression on the cell surface. We thus sought to investigate if this MED12 function distinct of the Mediator complex in TGFβ pathway activation may be involved in ATRi sensitivity. Co-depletion of TGFBR2 restored the VE822 sensitivity of MED12-knockdown cells to that of wildtype cells, in both cellular proliferation and clonogenic sensitivity assays (Figs 7A and 7B and S7). Similar results were obtained for AZD6738 sensitivity (S6E Fig). These results show that TGFBR2 is necessary for the ATRi resistance caused by MED12 depletion. To test if this involves TGFβ pathway activation, we measured the levels of phosphorylated SMAD2 (pSMAD2), a key mediator of TGFβ signaling. Depletion of MED12, but not of MED7 or of other CDK8 module subunits, resulted in

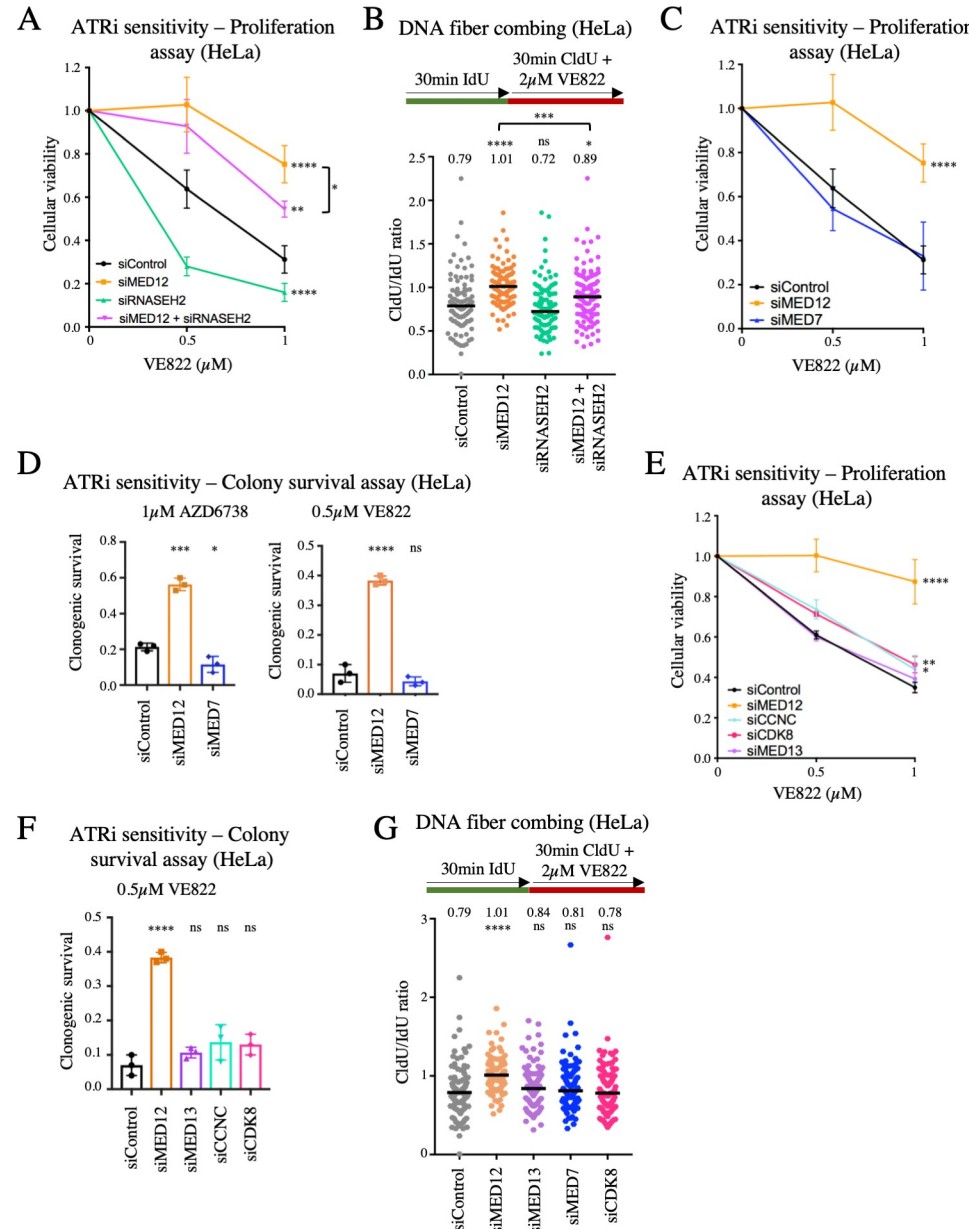

**Fig 6. The role of MED12 in the cellular response to ATR inhibitors does not involve the Mediator complex.** (**A**) Cellular proliferation assay showing that RNASEH2 depletion in HeLa cells results in VE822 sensitivity in both wildtype and MED12-knockdown cells. The average of four experiments is shown, with error bars representing standard deviations. Asterisks indicate statistical significance compared to the control sample, unless otherwise indicated. (**B**) DNA fiber combing experiment showing that RNASEH2 depletion reduced replication fork speed in both wildtype and MED12-knockdown HeLa cells, upon treatment with 2μM VE822. The median values are indicated for each sample, and the asterisks indicate statistical significance compared to the control sample, unless otherwise indicated. At least 100 fibers were quantified. (**C, D**) Unlike MED12 depletion, knockdown of the core Mediator complex subunit MED7 in HeLa cells does not promote ATRi resistance. Cellular proliferation (**C**) and colony survival (**D**) experiments are shown. The average of four (**C**) or three (**D**) experiments is shown, with error bars representing standard deviations. Asterisks indicate statistical significance compared to the control sample. (**E, F**) Unlike MED12 depletion, knockdown of CDK8 module subunits MED13, CCNC, or CDK8 in HeLa cells does not promote VE822 resistance. Cellular proliferation (**E**) and colony survival (**F**) experiments are shown. The average of three experiments is shown, with error bars representing standard deviations. Asterisks indicate statistical significance compared to the control sample. (**G**) DNA fiber combing experiment showing that, unlike MED12 depletion, knockdown of MED7, MED13 or CDK8 in HeLa cells does not restore normal fork speed upon treatment with 2μM VE822. The median values are indicated for each sample, and the asterisks indicate statistical significance compared to the control sample. At least 100 fibers were quantified.

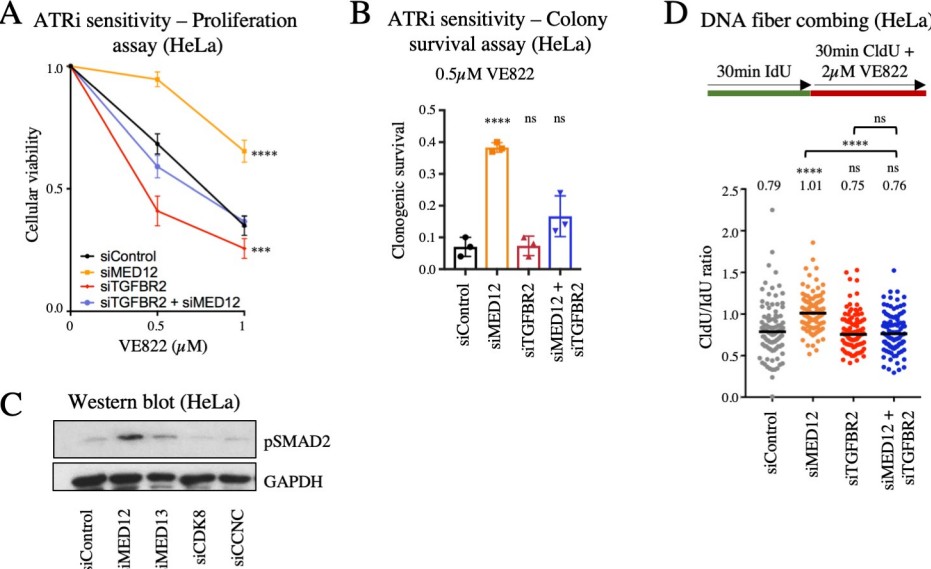

**Fig 7. Loss of MED12 promotes ATRi resistance through activation of the TGFβ pathway.** (**A**, **B**) TGFBR2 co-depletion restores VE822 sensitivity in MED12-knockdown HeLa cells. Cellular proliferation (**A**) and colony survival (**B**) experiments are shown. The average of three experiments is shown, with error bars representing standard deviations. Asterisks indicate statistical significance compared to control. (**C**) Western blot showing that MED12 depletion, but not knockdown of MED13, CDK8, or CCNC induces TGFβ pathway activation as measured by phosphorylation of SMAD2 in HeLa cells. (**D**) DNA fiber combing experiment showing that TGFBR2 co-depletion restores fork slowing induced by treatment with 2μM VE822 in MED12-knockdown HeLa cells. The median values are indicated for each sample, and the asterisks indicate statistical significance compared to the control sample, unless otherwise indicated. At least 100 fibers were quantified.

increased SMAD2 phosphorylation (Fig 7C), indicating hyper-activation of the TGFβ pathway.

Finally, we investigated if the restoration of replication fork speed in the presence of ATR inhibitors observed upon MED12 depletion also involves the TGFβ pathway. Similar to the ATRi sensitivity results, co-depletion of TGFBR2 suppressed the increased replication fork speed of MED12-knockdown cells to wildtype levels (Fig 7D). This indicates that activation of the TGFβ pathway upon MED12 depletion promotes the increased fork speed in the presence of ATRi treatment observed in these cells.

## Genome-wide CRISPR activation screens identify genes whose overexpression causes resistance to multiple ATRi

To evaluate genes whose overexpression leads to resistance to ATR inhibitors, we performed CRISPR activation screens in HeLa cells as well as MCF10A cells. The Calabrese CRISPR activation library was employed for these experiments [41]. This library targets 18,885 genes with 56,762 gRNAs, for an average of 3 guides per gene. First, HeLa and MCF10A cells were infected with a lentivirus containing the dCas9 construct necessary for transcriptional activation (S8A Fig). Expression of dCas9 was confirmed by Western blot (S8B Fig). These cells were then infected with the activation library. Next, 30 million library-infected cells (for 500-fold library coverage) were treated with ATRi using the same conditions as for the knockout screens (Fig 1). Surviving cells were collected and genomic DNA was extracted. The gRNA sequences were PCR-amplified and identified by Illumina sequencing (Fig 1B and 1C). Using the RSA algorithm, we generated separate lists of genes that were enriched in the VE822 and

AZD6738 conditions compared to the control (S3 Table). This represents genes that, when overexpressed, confer resistance to ATRi. Similar to the knockout screen, there was large overlap between the VE822 and AZD6738 conditions. Within the top hits of each of the two ATRi HeLa screens with logP values lower than -2.0 (241 genes for the VE822 screen and 235 genes for the AZD6738 screen), 50 were present in both of them (Fig 8A and S4 Table). Within the

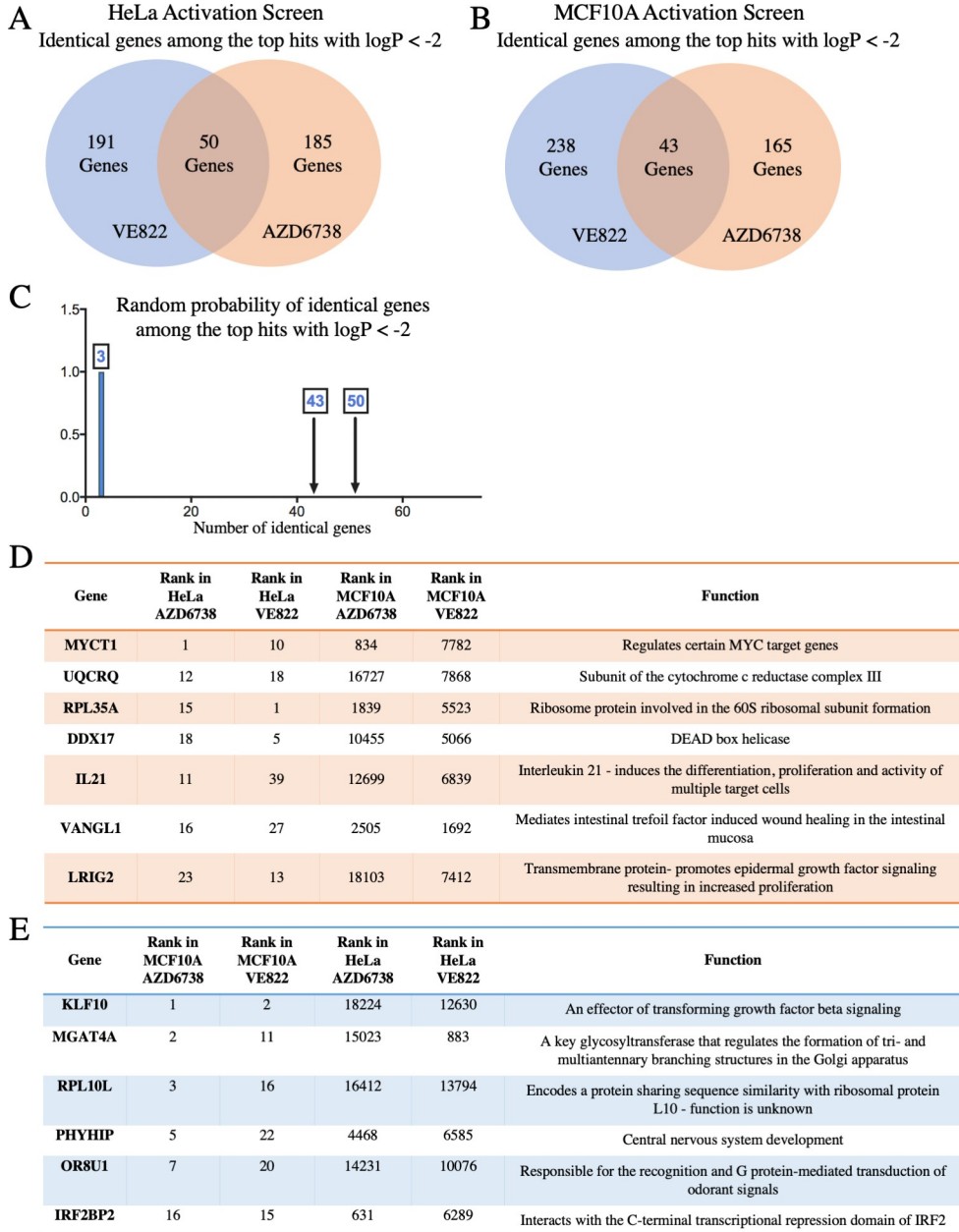

**Fig 8. CRISPR activation screens in HeLa and MCF10A cells to identify genes that result in resistance to multiple ATRi when overexpressed.** (**A**) Diagram showing the overlap of identical genes within the top hits of the ATRi activation screens in HeLa cells with logP values lower than -2.0. (**B**) Diagram showing the overlap of identical genes within the top hits of the ATRi activation screens in MCF10A cells with logP values lower than -2.0. (**C**) The number of common genes within the top hits with logP values lower than -2.0 (namely 50 for the HeLa screens and 43 for the MCF10A screens) is much higher than the random probability of identical hits. (**D, E**) Tables listing the common genes among top 40 hits in each of the ATRi CRISPR activation screens in HeLa (**D**) and MCF10A (**E**) cells.

top hits of the MCF10A screens with logP values lower than -2.0 (281 genes for the VE822 screen and 208 genes for the AZD6738 screen), 43 were present in both of them (Fig 8B and S4 Table). The number of common hits is much higher than expected from a random distribution (Fig 8C). Moreover, within the top 40 genes, 7 hits were common in HeLa cells (Figs 8D) and 6 were common in MCF10A cells (Fig 8E). Importantly, all seven top hits from the HeLa knockout screen, were ranked towards the bottom in the HeLa overexpression screens (Fig 9A), indicating that their overexpression promotes ATRi sensitivity and thus further highlighting the relevance of our screening strategy.

Interestingly, when comparing between the HeLa and MCF10A screens, there was no overlap of top hits (S3 Table). This surprising finding perhaps reflects the inherent differences between these cell lines, as one is cancer-derived and p53 deficient, while the other is non-transformed with wild-type p53 function. To confirm these findings, we chose to validate hits that were common to both ATR inhibitors, but cell line specific. RPL35A and MYCT1 were top hits for both ATRi screens in HeLa cells, but were not in the MCF10A screens (Fig 9A and 9B). On the other hand, IRF2BP2 and MGAT4A were top hits for both ATRi screens in the MCF10A cells, but not in the HeLa screens (Fig 9A and 9B). We overexpressed these four genes in the HeLa dCas9 cells by CRISPR activation using the MS2-P65-HSF1 (MPH) activator complex, to induce transcription. Western blots confirmed the overexpression of all four genes (S9 Fig). Cellular proliferation of the overexpression cell lines was analyzed upon incubation with VE822 for three days. Overexpression of RPL35A or of MYCT1 resulted in ATRi resistance compared to control cells, while overexpression of IRF2BP2 or of MGAT4A did not show any difference to control cells (Fig 9C). These results are in line with the screen data that identified RPL35A and MYCT1 as top hits in HeLa cells but not in MCF10A cells, thus validating the activation screens. These findings suggest that different mechanisms may regulate ATRi resistance induced by genes overexpression in HeLa compared to MCF10A cells.

## Discussion

Detailed information on the genetic make-up of tumors will help to better treat patients on a personalized basis. Identification of markers that lead to resistance to ATRi is critical to advance the use of ATRi in cancer therapy. With the emerging use of ATRi in clinical trials, there has been renewed interest in determining predictors to ATRi response. Moreover, beyond the treatment of patients, the effects of ATRi on cell biology and pathways that regulate responses to ATRi are still not well understood.

Here, we present dual-genome wide CRISPR knockout and activation screens to identify predictors of ATRi resistance in HeLa and MCF10A cells. Specifically, two of the main ATRi that are currently in clinical trials, VE822 and AZD6738 were investigated. Our CRISPR screening strategy provides an unbiased approach to identify genes that, when altered, cause resistance to ATRi. First, we performed a CRISPR knockout screen in HeLa cells to identify genes that, when knocked out, caused resistance to VE822 and AZD6738. When comparing the results for the two ATRi screens, we observed a significant overlap of the top hits, much higher than expected from a random distribution. This validates our screening strategy and shows that common pathways are involved in the response to different ATRi. We confirmed that loss of the seven top-ranked common gene hits causes resistance to both VE822 and AZD6738, in three different cell lines using siRNA knockdown. Mechanistically, we show that none of these cases involve restoration of ATR activity in the presence of the inhibitors. Instead, we identified different mechanisms through which some of these genes regulate ATRi resistance.

First, we show that regulation of apoptosis can affect the cellular response to ATRi. One of the top hits in our screen, namely LUC7L3, was previously involved in the regulation of the

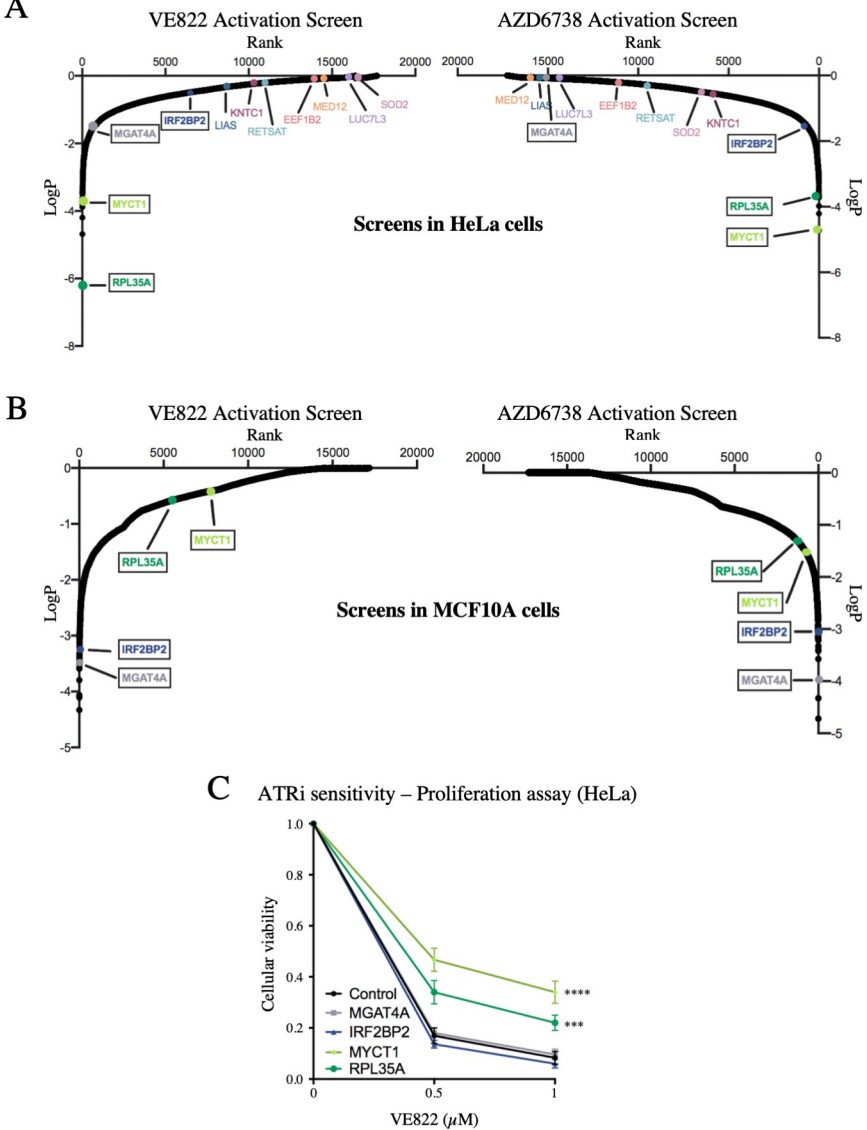

**Fig 9. Validation of the CRISPR activation screens for ATRi resistance.** (**A**) Scatterplot showing the results of the ATRi activation screens in HeLa cells. Each gene targeted by the library was ranked according to P-values calculated using RSA analysis. The P-values are based on the fold change of the guides targeting each gene between the ATRi- and DMSO-treated conditions. The location of RPL35A, MYCT1, IRF2BP2 and MGAT4A is shown (in black boxes). RPL35A and MYCT1 were top hits in both ATRi HeLa screens, whereas IRF2BP2 and MGAT4A were not. Also shown is the location of the seven top hits from the HeLa knockout screen described above. (**B**) Scatterplot showing the results of the ATRi activation screens in MCF10A cells. IRF2BP2 and MGAT4A were top hits in both ATRi screens, whereas RPL35A and MYCT1 were not. (**C**) Overexpression of RPL35A or MYCT1 in HeLa cells causes resistance to VE822 in a cellular proliferation assay, while overexpression of IRF2BP2 or MGAT4A does not. The average of three experiments is shown, with error bars representing standard deviations. Asterisks indicate statistical significance compared to control.

expression of the apoptotic regulator BCL-X through alternative splicing [30]. While the impact of LUC7L3 on BCL-X splicing was not directly investigated, it was proposed that its interaction with the RNA binding protein RBM25, which binds to the BCL-X pre-mRNA, is important for recruitment of the U1 snRNP complex to the BCL-X pre-mRNA to initiate alternative splicing from an anti-apoptotic to a pro-apoptotic form of BCL-X. This model is

confirmed by our studies, which directly tested the impact of LUC7L3 on BCL-X splicing upon ATRi treatment. We found that, upon loss of LUC7L3, the level of the pro-apoptotic iso-form of BCL-X decreases, resulting in reduced ability to undergo apoptosis, and thus poten-tially explaining the ATRi inhibitor resistance of these cells.

Another mechanism of resistance we uncovered involves restoration of replication fork pro-tection. It was previously shown that ATR inhibition causes a decrease in fork progression and an increase in origin firing [42]. Under normal conditions, ATR is responsible for suppression of local origin firing, therefore when ATR is inhibited, origin firing increases [43]. As there is an inverse correlation between DNA tract length and the number of origins firing, the more ori-gins that fire, the slower the fork progression rate is [44]. However, in the case of ATRi it is still unknown whether replication slowing causes more origins to fire, or if more origins firing cause the replication forks to slow [45]. We found that loss of either MED12 or LIAS causes a restoration of fork stability as the decrease in fork progression upon ATRi treatment is not seen under these conditions. This result suggests that mechanisms countering fork slowing or origin firing in the presence of ATRi, may in fact promote cellular viability under these conditions.

Surprisingly, we found that MED12 exerts its function in regulating ATRi sensitivity to a large extent in a manner independent of the Mediator complex. Instead, similar to the previously described role of MED12 in regulating sensitivity to inhibition of ALK and EGF receptor tyrosine kinases [40], we found that loss of MED12 results in ATRi resistance by activating the TGFβ path-way through increasing TGFBR2 expression. Importantly, hyper-activation of the TGFβ pathway upon MED12 depletion was also required for the increased replication fork speed upon ATRi treatment observed in these cells. How exactly TGFβ signaling promotes replication fork protec-tion is an important open question. The TGFβ pathway is involved in numerous cellular pro-cesses including cell cycle regulation, cell death, and cell migration [46]. In addition, the TGFβ pathway has been shown to modulate the expression of several DNA repair genes [47, 48], and be involved in regulation of double strand break repair [49–53]. It remains to be seen if these activi-ties of the TGFβ pathway also underlie its novel function in replication fork stability.

While our findings indicate a Mediator complex-independent role for MED12 in mediating the response to ATRi through regulation of the TGFβ pathway, it is possible that the Mediator complex also impacts ATRi sensitivity. Our findings that RNASEH2 depletion results in ATRi sensitivity in both wildtype and MED12-depleted cells argue against the model that MED12 loss promotes ATRi resistance by suppressing R-loop formation. However, this possibility can-not be excluded. Indeed, some R-loops may still form in cells deficient in the Mediator com-plex, which could explain why RNASEH2 depletion promotes ATRi sensitivity in MED12-deficient cells. In line with this, additional subunits of the CDK8 module of the Medi-ator complex were associated with resistance to ATRi inhibitors in previously published screens [25], suggesting a role of this complex in regulating the ATRi response, separate from the MED12-specific role in TGFβ pathway regulation. Since we have not observed a significant impact of other Mediator complex members on ATRi resistance in our studies, the impact of this complex may perhaps depend on the cell type.

Finally, we completed the first (to our knowledge) genome-wide CRISPR activation screen to identify genes that cause resistance to ATRi when overexpressed. This activation screen was performed in both HeLa and MCF10A cells. Validating the screening strategy, top hits from the knockout screen ranked very low on the list of hits for the activation screens. Similar to the HeLa knockout screen, there was a high number of identical top hits between the VE822 and AZD6738 screens for each cell line. However, there was almost no overlap between the two cell lines. We confirmed these surprising results by showing that overexpression of RPL35A or MYCT1, two top hits from the HeLa screens, causes ATRi resistance in HeLa cells, but overex-pression of IRF2BP2 or MGAT4A, top hits from the MCF10A screens, does not cause

resistance in HeLa cells. These findings may reflect the inherent differences between the two cell lines, as HeLa cells are tumor-derived, whereas MCF10A cells are non-transformed breast epithelial cells. Compared to knockout approaches, which lead to inactivation of genetic pathways, overexpression approaches result in amplification of pathways which may already be active in cells. The resulting phenotype may ultimately depend upon a complex interaction with other pathways and processes operating in the respective cells, and thus be affected to a much larger extent by the biology and transcriptional program of each cell type. Thus, our results suggest that biomarkers of gene overexpression specific to tumor cells may be used to create treatment plans with fewer side effects to the rest of the body.

For our screens, we treated cells for 108 hours with high doses of ATRi that killed 90% of cells. This combination of a high ATRi concentration and a relatively short treatment time may potentially bias towards identification of genes that regulate the acute apoptotic response. However, we found that the loss of the top hits identified also restore ATRi resistance in long-term clonogenic assays, confirming that the screens identified true regulators of ATRi resistance. Nevertheless, longer treatment with a lower drug concentration may have identified additional mechanisms. In the case of PARP inhibitors, cell death was shown to require progression through mitosis, as the drugs induce chromosomal abnormalities which result in mitotic catastrophe [45, 54]. It is likely that such mechanisms would still be uncovered by our screens, since the treatment time we employed allowed for several cell divisions to occur.

Our studies employed a unique combination of genome-wide CRISPR-based screening approaches to comprehensively identify genes that, when altered, cause resistance to ATRi. We analyzed two separate ATRi using both knockout and activation screens, in both cancer cells and non-transformed cells. Our findings could provide biomarkers to ultimately help create effective treatment plans for cancer therapy with ATR inhibitors.

## Materials and methods

### Cell culture

HeLa and 8988T cells were grown in Dulbecco's modified Eagle's medium (DMEM) supplemented with 10% fetal calf serum and 1% Pen/Strep. MCF10A cells were grown in DMEM/F12 supplemented with 5% fetal calf serum, 1% Pen/Strep, 20ng/mL hEGF, 0.5 mg/mL Hydrocortisone, 100 ng/mL Cholera Toxin, and 10 μg/mL Insulin.

Gene knockdown was performed using Lipofectamine RNAiMAX transfection reagent. Cells were treated with siRNA for two consecutive days. The following SilencerSelect oligonucleotides (ThermoFisher) were used for gene knockdown: KNTC1 (ID: s18776); LUC7L3 (ID: s226748); SOD2 (ID: s13267); LIAS (ID: s223178); EEF1B2 (ID: s194388); RETSAT (ID: s29671); MED12 (ID: s19362); MED7 (ID: s18080), MED13 (ID: s19365), CCNC (ID: s391), CDK8 (ID: s2831), TGFBR2 (ID: s14077), RNASEH2 (ID: s20656), RBM25 (ID: s33912). AllStars Negative Control siRNA (Qiagen 1027281) was used as control.

HeLa cells overexpressing RPL35A, IRF2BP2, MYCT1 or MGAT4A were created by consecutive rounds of transduction and selection to induce transcriptional activation. Cells were first transduced with the dCas9 lentiviral construct (Addgene 61425-LV) and selected with 3μg/ml blasticidin. The resulting HeLa-dCas9 cells were then transduced with the lentiviral construct for the MS2-P65-HSF1 (MPH) activator complex (Addgene 61426-LVC) and selected with 0.5 mg/ml hygromycin. Finally, HeLa-dCas9-MPH cells were transduced with lentivirus constructs containing the following guide sequences: CGGTGGCGGCCGCGTCCCGG for IRF2BP2, CAGTGCGAAGCCGATTTCCG for RPL35A, AGAACTTAGGAACTTAGCTG for MYCT1, and CCAGCCATTGGCCGGCCCCG for MGAT4A (Sigma-Aldrich Custom CRISPR in lentiviral backbone LV06).

## Protein techniques

Cell extracts and western blots were performed as previously described [55]. Antibodies used were: MED12 (Santa Cruz Biotechnology sc-515695), SOD2 (Santa Cruz Biotechnology sc-133254), LUC7L3 (ProteinTech 145041AP), EEF1B2 (ProteinTech 104831AP), BCL-X$_{short}$ (Invitrogen PA5-78864), Vinculin (Santa Cruz Biotechnology sc-73614), GAPDH (Santa Cruz Biotechnology sc-47724), Cas9 (BioLegend 844302), pChk1-S317(Cell Signaling Technology 2344S), pChk1-S345(Cell Signaling 2341S), IRF2BP2 (ProteinTech 188471AP), RPL35A (Bethyl, 501569555), MED7 (Abcam ab187146), MED13 (Invitrogen PA5-35924), CCNC (Bethyl A301-989A-T), CDK8 (ProteinTech 220671AP), TGFBR2 (Cell Signaling Technology 79424S), RNASEH2 (Bethyl A304-149A-T), RBM25 (Abcam ab72237), γH2AX (Bethyl A300-081A), phospho-SMAD2 Ser465/Ser467 (Cell Signaling Technology 18338T), MGAT4A (Abcam ab151750), MYCT1 (Abcam ab139945), BCL-X$_{long}$, (Santa Cruz Biotechnology sc-271121), RETSAT (invitrogen PA5-65443), KNTC1 (Bethyl A301-712A-T), LIAS (Abcam ab96302).

## CRISPR screens

For CRISPR knockout screens, the Brunello Human CRISPR knockout pooled lentiviral library (Addgene 73179) was used [26]. This library is comprised of 76,411 gRNAs that target 19,114 genes. Fifty million HeLa cells were infected with this library at a multiplicity of infection (MOI) of 0.4 to achieve 250-fold coverage and selected for 4 days with 0.6 μg/mL puromycin. For ATRi resistance screens, 20 million library-infected cells (to maintain 250-fold coverage) were used for each drug condition: DMSO (vehicle control), 1.5μM VE822 (Selleck S7102), and 3.6μM AZD6738 (Selleck S7693). Cells were treated for 108 hours and then collected. Compared to control cells, survival of ATRi-treated cells was 8% (for VE822) and 10% (for AZD6738) respectively.

For CRISPR activation screens, the Calabrese Human CRISPR activation pooled library, targeting 18,885 genes with 56,762 gRNAs, was used (Set A, AddGene 92379) [41]. First, wild-type HeLa and MCF10A cells were infected with the dCas9 lentiviral construct (Addgene 61425-LV) and selected with 3μg/ml blasticidin for 5 days. The presence of the dCas9 was confirmed by western blot using a Cas9 antibody. Next, 75 million HeLa-dCas9 and MCF10A-dCas9 cells were infected with this library at a MOI of 0.4 to achieve 500-fold coverage and selected for 96 hours with 0.6 μg/mL puromycin. For ATRi resistance screens, 30 million cells were used for each drug treatment condition to maintain 500-fold coverage of the library. Library-infected HeLa-dCas9 and MCF10A-dCas9 cells were treated with DMSO (vehicle control), 1.5μM VE822, or 3.6μM AZD6738 for 108 hours. For HeLa cells, survival of ATRi treated cells was 9% (VE822) and 11% (AZD6738), respectively. For MCF10A cells, survival was 15% (VE822) and 13% (AZD6738), respectively.

## Sequencing and analysis of CRISPR screens

Genomic DNA was isolated using the DNeasy Blood and Tissue Kit (Qiagen 69504) per the manufacturer's instructions. gRNAs were identified in our populations using PCR primers with Illumina adapters. Genomic DNA from a number of cells corresponding to the equivalent of 250-fold library coverage was used as template for PCR (20 million cells for knockout screen, 15 million cells for activation screen). 10μg of gDNA was used in each PCR reaction along with 20μl 5X HiFi Reaction Buffer, 4μl of P5 primer, 4μl of P7 primer, 3μl of Radiant HiFi Ultra Polymerase (Stellar Scientific), and water. The P5 and P7 primers were determined using the user guide provided with the CRISPR libraries (https://media.addgene.org/cms/filer_public/61/16/611619f4-0926-4a07-b5c7-e286a8ecf7f5/broadgpp-sequencing-protocol.

pdf). The PCR cycled as follows: 98˚C for 2min before cycling, then 98˚C for 10sec, 60˚C for 15sec, and 72˚C for 45sec, for 30 cycles, and finally 72˚C for 5min. After PCR purification, the final product was Sanger sequenced to confirm that the guide region is present, followed by qPCR to determine the exact amount of PCR product present. The purified PCR product was then sequenced with Illumina HiSeq 2500 single read for 50 cycles, targeting 10 million reads.

Next, the sequencing results were analyzed bioinformatically. First, the gRNA representation was analyzed using the custom python script provided (count_spacers.py). [56]. The difference between the number of guides present in each ATRi condition compared to control condition was then determined. Specifically, one read count was added to each gRNA, and then the treatment reads were normalized to no treatment. Finally, the values found were used as input in the Redundant siRNA Activity (RSA) algorithm [27, 57]. For RSA, the Bonferroni option was used and guides that were 2-fold enriched in treatment compared to no treatment were considered hits. This analysis method allows for the identification of genes that are upregulated in one population (VE822 or AZD6738) compared to control. Hits are determined by the amount of gRNA sequences present in the population and the number of guides per gene present. Furthermore, p-values are determined by the RSA algorithm for the genes that are most enriched in the test populations compared to the control. VE822 and AZD6738 top hits can then be compared. Biological pathway analysis of the top hits was performed using Gene Ontology [58, 59].

### Drug sensitivity assays

Cellular proliferation was measured using the CellTiterGlo reagent (Promega) according to the manufacturer's instructions. After 2 days of siRNA treatment, 1500 cells per condition were plated into 96-well plates and treated with the indicated doses of ATRi for three days. CellTiterGlo reagent was added for 10 minutes before the luminescence was read on a plate reader. For colony survival assays, 500 siRNA treated cells were plated into 6-well plates and treated with the indicated doses of ATRi. After 3 days of treatment, media was replaced. After two weeks, cells were fixed and colonies were stained with crystal violet. Apoptosis was quantified using the FITC Annexin V kit (Biolegend 640906) according to the manufacturer's instructions.

### DNA fiber combing assay

HeLa cells were treated with siRNA as indicated, then incubated with 100μM IdU in DMEM for 30 minutes. Cells were then washed three times with PBS and incubated with 100μM CldU and VE822 as indicated, in DMEM for 30 minutes. Next, cells were collected and processed using the FiberPrep kit (Genomic Vision EXT-001) according to the manufacturer's instructions. DNA molecules were stretched onto coverslips (Genomic Vision COV-002-RUO) using the FiberComb Molecular Combing instrument (Genomic Vision MCS-001). The slides were then stained with antibodies detecting CldU (Abcam 6236), IdU (BD 347580), and DNA (Millipore Sigma MAB3034). Next, slides were incubated with secondary Cy3, Cy5, or BV480-conjugated antibodies (Abcam 6946, Abcam 6565, and BD Biosciences 564879). Finally, the cells were mounted onto coverslips and imaged using confocal microscopy (Leica SP5).

### Reverse transcriptase PCR (RT-PCR)

To detect the long and short isoforms of BCL-X mRNA, RT-PCR was used. RNA was extracted using TRIzol reagent (Life Tech) according to the manufacturer's instructions. RNA was then converted to cDNA using the RevertAid Reverse Transcriptase Kit (Thermo Fisher Scientific)

with oligo-dT primers. Next, PCR was performed with the following primers: GAPDH (for: TGCACCACCAACTGCTTAGC; rev: TCAGCTCAGGGATGACCTTG), BCL-X set 1 (for: AGTAAAGCAAGCGCTGAGGGAG; rev: ACTGAAGAGTGAGCCCAGCAGA) [60], BCL-X set 2 (for: GAGGCAGGCGACGAGTTTGAA; rev: TGGGAGGGTAGAGTGGATGGT) [61]. The PCR cycled as follows: 95˚C for 2min to start, followed by 95˚C for 30sec, 45˚C for 1min, and 68˚C for 1min (cycled 30 times), with a final 5-minute incubation at 68˚C. The PCR product was then run on a 2% agarose gel and imaged with a ChemiDoc Gel Imager (Bio-Rad).

## Statistical analyses

The Mann-Whitney statistical test was performed for the DNA fiber assay. The CellTiterGlo proliferation assays were analyzed with a 2-way ANOVA. For other assays, the t-test (two-tailed, unequal variance unless indicated) was performed. Statistical significance is indicated for each graph (ns = not significant, for $P > 0.05$; * for $P \leq 0.05$; ** for $P \leq 0.01$; *** for $P \leq 0.001$; **** for $P \leq 0.0001$). The random probabilities of identical genes within the top hits with logP values lower than -2.0 were calculated by multiplying the individual probabilities of each set: [(number of genes in set 1/total number of genes in the library) * (number of genes in set 2/total number of genes in the library)].

## Data availability

All source data underlying each of the figures, including the values plotted in graphs, the exact p-values, and the uncropped blots are presented in S5 Table (for the main figures) and S6 Table (for the supplemental figures).

## Supporting information

**S1 Fig. Validation of the siRNA-mediated depletion of the top hits.** Western blots showing knockdown of SOD2 (**A**), MED12 (**B**), LUC7L3 (**C**), EEF1B2 (**D**), RETSAT (**E**), KNTC (**F**), and LIAS (**G**) in HeLa cells are presented.
(TIF)

**S2 Fig. Levels of pChk1 are not restored upon knockdown of the top hits and subsequent treatment with ATRi.** Western blots showing the levels of pCHK1 S317 (**A**) and pChk1 S345 (**B**) in HeLa cells after knockdown of the top hits followed by 24 hours of no treatment, hydroxyurea treatment, ATRi treatment, or hydroxyurea with ATRi treatment, are shown. (**C**) Western blots showing no impact on γH2AX levels upon knockdown of the top hits in HeLa cells.
(TIF)

**S3 Fig. The LUC7L3-RBM25 complex regulates ATRi resistance through BCL-X splicing.** (**A**) Reverse transcriptase PCR showing a decrease in BCL-X$_{short}$ mRNA after knockdown of LUC7L3 followed by ATRi treatment in HeLa cells. Two different primer sets for BCL-X$_{short}$ were used, and GAPDH was used as control. (**B**) Western blots showing RBM25 depletion by siRNA-mediated knockdown in HeLa cells. (**C**) Western blot showing an increase in the protein levels of the BCL-X$_{long}$ isoform in HeLa cells after knockdown of LUC7L3 or RBM25 and ATRi treatment.
(TIF)

**S4 Fig. Validation of siRNA-mediated depletion of RNASEH2.** Western blots showing the siRNA-mediated co-depletion of RNASEH2 and MED12 in HeLa cells are presented.
(TIF)

**S5 Fig. Impact of transcription inhibition on ATRi sensitivity.** Cellular proliferation experiment showing that inhibition of RNA polymerase II using α-amanitin does not impact VE822 sensitivity of HeLa cells. Cells were treated with both drugs for 3 days at the indicated concentrations. The average of four experiments is shown, with error bars representing standard deviations.
(TIF)

**S6 Fig. Impact of Mediator subunits and the TGFβ pathway on ATRi resistance.** (**A-D**) Western blots showing knockdown of MED7 (**A**), MED13 (**B**), CCNC (**C**), and CDK8 (**D**) in HeLa cells are presented. (**E**) Cellular proliferation experiment showing the AZD6738 sensitivity of HeLa cells upon depletion of MED12, CDK8. CCNC, or TGFBR2. The average of three experiments is shown, with error bars representing standard deviations. Asterisks indicate statistical significance.
(TIF)

**S7 Fig. Validation of siRNA-mediated depletion of TGFBR2.** Western blots showing the siRNA-mediated co-depletion of TGFBR2 and MED12 in HeLa cells are presented.
(TIF)

**S8 Fig. Overview of the CRISPR activation screen.** (**A**) Schematic representation of the CRISPR activation screen setup. The gRNA targets dCas9 to the promoter region of the gene of interest, along with multiple transcriptional activators to upregulate the transcription of the gene. (**B**) Western blots showing dCas9 expression in the cells used for the CRISPR activation screen and HeLa-dCas9 overexpression cell lines.
(TIF)

**S9 Fig. Validation of the overexpression of the activation screen hits investigated.** Western blots showing the overexpression of IRF2BP2 (**A**), RPL35A (**B**), MYCT1 (**C**) and MGAT4A (**D**) in HeLa cells are presented.
(TIF)

**S1 Table. Lists of all genes in the CRISPR knockout screens ranked by p-value.**
(XLSX)

**S2 Table. List of common hits from the knockout screen.**
(XLSX)

**S3 Table. Lists of all genes in the CRISPR activation screens ranked by p-value.**
(XLSX)

**S4 Table. List of common hits from the activation screens.**
(XLSX)

**S5 Table. The source data underlying each of the main figure panels, including: The values plotted in graphs, the exact p-values, and the uncropped blots.**
(XLSX)

**S6 Table. The source data underlying each of the supplemental figure panels, including: The values plotted in graphs, the exact p-values, and the uncropped blots.**
(XLSX)

## Acknowledgments

We would like to thank Dr. Hong-Gang Wang and Dr. Tom Spratt for materials and advice; and the following Penn State College of Medicine core facilities: Flow Cytometry,

Genomic Analyses, and Imaging. Experimental design schemes and models were created with Biorender.com.

## Author Contributions

**Conceptualization:** Emily M. Schleicher, George-Lucian Moldovan.

**Data curation:** Emily M. Schleicher, Ashna Dhoonmoon, Lindsey M. Jackson, Kristen E. Clements, Coryn L. Stump, Claudia M. Nicolae.

**Formal analysis:** Emily M. Schleicher, Ashna Dhoonmoon, Lindsey M. Jackson, Kristen E. Clements, Coryn L. Stump, Claudia M. Nicolae.

**Funding acquisition:** Emily M. Schleicher, George-Lucian Moldovan.

**Methodology:** Emily M. Schleicher, Kristen E. Clements.

**Supervision:** George-Lucian Moldovan.

**Validation:** Emily M. Schleicher, Claudia M. Nicolae.

**Writing – original draft:** Emily M. Schleicher, George-Lucian Moldovan.

**Writing – review & editing:** Emily M. Schleicher, George-Lucian Moldovan.

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
