## [Decision Letter · Decision Letter 0]

13 May 2020

Dear Dr Moldovan,

Thank you very much for submitting your Research Article entitled 'Dual genome-wide CRISPR knockout and CRISPR activation screens identify common mechanisms that regulate the resistance to multiple ATR inhibitors' to PLOS Genetics. Your manuscript was fully evaluated at the editorial level and by independent peer reviewers. The reviewers appreciated the attention to an important problem, but raised some substantial concerns about the current manuscript, including suggestions of additional experiments to bring the paper on par with PLOS Genetics expectations of advance and novelty.  Based on the reviews, we will not be able to accept this version of the manuscript, but we would be willing to review again a much-revised version. We cannot, of course, promise publication at that time.

If you decide to revise the manuscript for further consideration at PLOS Genetics, please aim to resubmit within the next 60 days.  We understand that many labs are completely or partially closed at this time, so extension over suggested 60 days would be not unusual.  If you anticipate that  it will take extra time to address the concerns of the reviewers, we would appreciate an expected resubmission date by email to plosgenetics@plos.org.  

[LINK]

We are sorry that we cannot be more positive about your manuscript at this stage. Please do not hesitate to contact us if you have any concerns or questions.

Yours sincerely,

Dmitry A. Gordenin

Associate Editor

PLOS Genetics

Gregory Barsh

Editor-in-Chief

PLOS Genetics

Reviewer's Responses to Questions

**Comments to the Authors:**

Reviewer #1: This is a review of PGENETICS-D-20-00517, "Dual genome-wide CRISPR knockout and CRISPR activation screens identify common mechanisms that regulate the resistance to multiple ATR inhibitors." In this manuscript, the authors provide screening results for genes that cause resistance to ATR inhibitors, which are agents actively being pursued in cancer clinical trials. A thoughtful knockout screen is performed in HeLa cells with two independent ATR inhibitors, which uncovers 7 common genes among the top 50 hits. These 7 genes are used for several validation experiments, including in different cell types. This is perhaps the most interesting part of the paper, because depletion of none of them cause reactivation of ATR activity per se, some affect apoptosis, one is confirmed to directly affect apoptotic signaling, and two are shown to restore DNA replication fork speed. Accordingly, distinct mechanisms of resistance are raised, and new genes involved in these processes are uncovered. However, I would like to see a summary table as a main Table (not supplemental) of all these 7 genes & all the effects tested, including the effects that are common among them (like a "-" for reactivates ATR) (repeated below as point 1). A CRISPR activation screen is also shown, but followed-up to a lesser extent, although the main point is that HeLa and MCF10A are so different, which is important. Although, given that the activation screen is less developed, I wonder if the title could be more focused on the broad conclusion of "distinct mechanisms mediate ATR-inhibitor resistance" or something, since I'm not seeing that the two types of screening found "common mechanisms (repeated below as point 2). I have other editorial requests for clarification listed below. With the CRISPR activation and downstream analysis of the hits, the study is somewhat underdeveloped, but nonetheless, I think the data are solid and rigorous and will stimulate research in the area, and hence is an important advance in understanding the genetic pathways that affect cellular resistance to ATR inhibitors.

Points:

1. I request generating a summary table as a main Table (not supplemental) of all these 7 genes & all the effects tested, including the effects that are common among them (like a "-" for reactivates ATR). Include those phenotypes tested in multiple cell types. The findings in each cell of the table could either be quantitative with fold-effects or a qualitative, depending on what the authors determine is most rigorous/representative.

2. The title could be more focused on the broad conclusion of "distinct mechanisms mediate ATR-inhibitor resistance" or something, since I'm not seeing that the two types of screening found "common mechanisms."

3. The figure labeling could have a bit more information. In figure panels themselves, I request that the authors state the name of the cell line. If the same cell line is used in all the panels, then naming it in the "A" panel would be sufficient. For panels, like 2A, it can be useful to state what we are looking at, like "Gene knockouts causing increased fitness in HeLa."

4. Regarding interpretation, I think it is important to have a Discussion paragraph that describes the pros/cons of treating cells with a high-dose for a relatively short period of time. Namely, examining a higher dose over a shorter period of culturing may bias the screen towards pathways affecting apoptotic death, whereas a lower dose over a longer period of culturing may bias the screen towards pathways affecting chromosomal instability that causes death by mitotic catastrophe. This is a recurrent theme with PARP inhibitors, and so that literature may help with the paragraph, like this paper PMC5791853, but I am not specifically requesting the citation, this is just an example of the literature I'm talking about. The clonogenic survival validation assays are helpful in this extent (i.e. the top genes affect both acute sensitivity and long-term sensitivity), but this should be discussed.

Reviewer #2: To identify genes whose depletion or overexpression might give rise to resistance to ATR inhibition in tumour cells, Schleicher et al. have conducted CRISPR knock-out as well as CRISPR-activation screens using two different ATR inhibitors (ATRi). In contrast to previously published work involving ATRi, the three main novel angles of the current study are as follows: 1) rather than focusing on sensitization of the cells to ATRi, this work aims at revealing possible mechanisms of resistance to ATRi, which is an important but currently understudied aspect of cancer therapy; 2) the validity of the screen hits is enhanced by the fact that they came up as high-scoring in the screens using two different ATRi; 3) finally, the application of the CRISPRa strategy is also a relatively novel approach. Following CRISPR KO screens using two ATR inhibitors, Schleicher and colleagues focused on the 7 hits, which are common to the lists of 40 top hits in each screen. The authors do a thorough job at validating these hits in cell proliferation and clonogenic survival assays not only in HeLa cells, the cell line used for original screens, but in two additional cells lines. Even though the authors validate most of the hits from CRISPR KO screens in cell survival assays against ATRi, they do not provide sufficient insight into the actual mechanisms behind tumour resistance to ATR inhibition upon loss of these genes. Moreover, the outcomes of the CRISPRa screen, which is the most novel part of the manuscript, appear to be validated at best superficially.

Specific comments:

1. For the top seven hits identified in CRISPR KO screens, the common mechanism of ATRi resistance is reduction in apoptosis (Fig. 4B). However, only one of these genes, LUC7L3, appears to have a possible direct link to controlling apoptosis. As LUC7L3 had previously been shown to interact with RBM25 and the latter controls expression of the pro-apoptotic form of BCL-X, the authors tested if loss of LUC7L3 would result in decrease of pro-apoptotic BCL-X and they saw exactly that (Fig. 4D). But what is the molecular mechanism of LUC7L3 in RBM25/BCL-X control? E.g., does LUC7L3 affect stability and/or expression levels of RBM25? Does LUC7L3 re-expression in depleted cells complement for the loss of the protein? Regarding the rest of the hits affecting apoptosis, analysing transcriptomes of the depleted cells could have shed light on the changes in gene expression and provided possible links to regulation of cell fate decisions.

2. The authors demonstrate that loss of either MED12 or LIAS appears to stabilize replication fork (RF). But this is just an observation without any mechanistic insight as to how the RF stabilization is achieved. Appreciably, the LIAS function in this process might be very difficult to dissect, however, there are some obvious avenues to explore for MED12. Given that MED12 is part of a large complex involved in transcription regulation, it would make sense to test 1) if loss of other subunits of the Mediator complex would have the same effect on RF stability – even if they didn’t score as high as MED12 in the screen; 2) if the cross-talk between transcription and replication is important for the observed effect.

3. To be able to draw a firm conclusion that CRISPR activation screen has actually worked, the authors would have to validate more than one hit per cell line.

In summary, this study identifies genes whose loss or overexpression makes cells more resistant to ATRi. However, apart from establishing that resistance to ATRi might arise via different molecular pathways, the authors do not provide detailed mechanistic explanation of the resistance. Therefore, unless further mechanistic insight is provided for the top hits identified in the screens, I cannot recommend current manuscript for publication in PLOS Genetics.

Reviewer #3: In this study, the authors perform dual genome-wide CRISPR-based knockout and activation screens to identify genes that regulate resistance to ATR inhibitors. The authors performed both screens using two different ATR inhibitors, VE822 and AZD6738. The knockdown screen was performed in HeLa cells and the top hits were validated in two additional cells lines, MCF10A and 8988T. The CRISPR-activator screen was performed in HeLa and MCF10A cells. Given the clinical relevance of ATR inhibitors as chemotherapeutic agents, the aim of study is important and clinically significant. The conclusions are premature and not totally supported by data provided. The purpose of including data from both, knockout and activation screens is unclear, especially because the activation library screen validation is very superficial. The activation screen does not necessarily complement or strengthen the findings of the knockout screen. It may be better to describe only on the knockout screen and focus on one pathway (e.g. replication or apoptosis) to provide a mechanistic insight into ATRi resistance. At present, it seems to be like a list of pathways and genes involved in ATRi resistance without any clear understanding.

Other concerns and suggestions to strengthen the manuscript are listed below.

1) The authors should incorporate the dataset from published ATR inhibitor CRISPR knockout screens e.g. Hustedt et al, 2020 (https://royalsocietypublishing.org/doi/10.1098/rsob.190156). This reference should be included in the paper. This will represent a conserved set of genes and should improve the overall confidence of identifying genes for downstream analysis.

2) Fig 2A, what was the reason for selecting top 500 genes for the Venn Diagram? A cut-off to include genes in the Venn Diagram should be based on their functional significance or logP value. Including genes that do not have a significant effect is not meaningful.

3) Fig 2B: The algorithm used to calculate random probability of finding common hits should be clearly explained.

4) Fig 3D-F. The conclusions drawn from the data shown in these figures is not convincing. Clonogenic survival assay results shown in Fig 3C are clear and convincing. Similar assay should be performed for validation of the seven top candidates in other cell lines.

5) Fig 4B: ATR inhibitors induce DNA damage. Knockdown of 7 genes have been shown to reduce fraction of apoptotic cells, the authors should comment on DNA damage status by performing gamma-H2AX/53BP1 staining after knockdown.

6) Fig 4: RT- PCR showing a decrease in BCL-X short mRNA in response to LUC7K3 knockdown should include BCL-X long form as a control. What does set 1 and set 2 mean? If they represent two different experimental sets, then why only one GAPDH control?

7) Fig 6: The activator screen is very preliminary with not very convincing validation data (results shown in Fig 7E are not convincing) or mechanistic insights. Why only two genes identified in activation screen in HeLa cells were validated but none in MCF10A cells screening. The argument to explain the lack of overlap of genes identified in the two cell lines that ATRi resistance will be different in Hela and MCF10A is contradicted by a conserved response that was observed in the knockout screen genes.

8) Knockdown results of only four genes are shown in Suppl. Fig 1. If good antibodies are not available, qRT-PCR should be performed. Showing effect on ATRi resistance without the knockdown data is not very helpful.

9) Suppl. Fig. 2A. The GAPDH Western blot for siLIAS should be replaced with a better control blot.

**Have all data underlying the figures and results presented in the manuscript been provided?**

Reviewer #1: Yes

Reviewer #2: Yes

Reviewer #3: Yes

PLOS authors have the option to publish the peer review history of their article (what does this mean?). If published, this will include your full peer review and any attached files.

Reviewer #1: No

Reviewer #2: No

Reviewer #3: No

---

## [Decision Letter · Decision Letter 1]

28 Sep 2020

Dear Dr Moldovan,

Thank you very much for submitting your Research Article entitled 'Dual genome-wide CRISPR knockout and CRISPR activation screens identify mechanisms that regulate the resistance to multiple ATR inhibitors' to PLOS Genetics. Your manuscript was fully evaluated at the editorial level and by independent peer reviewers. The reviewers appreciated the attention to an important topic but identified some aspects of the manuscript that should be improved.  Specifically, reviewer 2 recommended to either perform some additional experiments or to modify conclusions/discussion to have all statements fully supported by data on their view.  We hope you will be able to address these concerns in revised version.

We therefore ask you to modify the manuscript according to the review recommendations before we can consider your manuscript for acceptance. Your revisions should address the specific points made by each reviewer.

[LINK]

Yours sincerely,

Dmitry A. Gordenin

Associate Editor

PLOS Genetics

Gregory Barsh

Editor-in-Chief

PLOS Genetics

Reviewer's Responses to Questions

**Comments to the Authors:**

Reviewer #1: This is a review of revised manuscript, PGENETICS-D-20-00517R1, "Dual genome-wide CRISPR knockout and CRISPR activation screens identify mechanisms that regulate the resistance to multiple ATR inhibitors." In my original review, I requested clarification edits largely to the Figures, and some expansion/clarifications to the Discussion. These revisions have been performed with excellence. I found the systems approach, combined with the new findings resulting from examining the genetics of ATRi resistance, and systematic validation / examination of key cell biology end points of the top hits to be a valuable contribution to the field. I appreciate the recommendations of the other reviewers to provide more depth in the analysis for a couple of the genes, which in my opinion have been addressed well with the new experiments on MED12 and LUC7L3. In short, my concerns have been adequately addressed.

Reviewer #2: Schleicher and colleagues have addressed most of the previously raised points and their revised manuscript is significantly improved. However, a couple of issues remain and need to be resolved/addressed before I could recommend this manuscript for publication.

1. In my view, including activation screens into this manuscript does not add much value to it. Even though the authors have validated a couple of additional hits in viability assays, the lack of mechanistic insight into why overexpression of those proteins modulates cellular responses to ATR inhibition, remains to be elucidated. Thus, it would make sense to leave the activation screens out and just focus on the knock-out screens. That said, in Figure 1C, the last two diagrams depicting activation screens in HeLa and MCF10A are identical and, in the future, should be contracted into one.

2. Regarding the mechanism of ATRi resistance caused by loss of MED12, the authors’ interpretation of the data in Fig. 6A&B, appears to be dubious. The claim that RNASEH2 depletion significantly reduces ATRi resistance (Fig. 6A) and RF speed (Fig. 6B) in MED12-depleted cells is hard to accept, given how minor the differences are. These data do not rule out that MED12 loss suppresses R-loop formation. In fact, it could be interpreted in a way that since loss of MED12 might suppress R-loop formation, the concomitant loss of RNASEH2 has only a minor effect on RF speed and viability in the presence of ATRi in MED12-depleted cells. As MED12 is not the only contributor to R-loop formation/resolution, in its absence, some R-loops can still form; hence, the residual effect of siRNASEH2 in MED12-depleted cells. To unequivocally the issue, the authors would have to experimentally test for abundance of RNA:DNA hybrids in their various genetic backgrounds. If those experiments are not conducted, however, I suggest that the authors at least modify their interpretation of the data in order not to exclude a very plausible hypothesis.

3. Point to add to discussion. The authors have tested the additional components of the Mediator complex in HeLa cells and their data show convincingly that MED12 has a separate function in mediating ATRi responses. Given that the Mediator proteins such as MED13, CDK8, CCNC have previously come up in ATRi screens together with MED12 (e.g. Wang et al., 2019, Oncogene), is it possible that the effects seen here are cell line specific to HeLa cells?

Reviewer #3: The authors have very satisfactorily addressed all my concerns.

**Have all data underlying the figures and results presented in the manuscript been provided?**

Reviewer #1: Yes

Reviewer #2: Yes

Reviewer #3: None

PLOS authors have the option to publish the peer review history of their article (what does this mean?). If published, this will include your full peer review and any attached files.

Reviewer #1: No

Reviewer #2: No

Reviewer #3: No

---

## [Editor Report · Decision Letter 2]

2 Oct 2020

Dear Dr Moldovan,

We are pleased to inform you that your manuscript entitled "Dual genome-wide CRISPR knockout and CRISPR activation screens identify mechanisms that regulate the resistance to multiple ATR inhibitors" has been editorially accepted for publication in PLOS Genetics. Congratulations!

Yours sincerely,

Dmitry A. Gordenin

Associate Editor

PLOS Genetics

Gregory Barsh

Editor-in-Chief

PLOS Genetics

Comments from the reviewers (if applicable):

**Data Deposition**

http://datadryad.org/submit?journalID=pgenetics&manu=PGENETICS-D-20-00517R2

**Press Queries**

---

## [Editor Report · Acceptance letter]

20 Oct 2020

PGENETICS-D-20-00517R2 

Dual genome-wide CRISPR knockout and CRISPR activation screens identify mechanisms that regulate the resistance to multiple ATR inhibitors 

Dear Dr Moldovan, 

We are pleased to inform you that your manuscript entitled "Dual genome-wide CRISPR knockout and CRISPR activation screens identify mechanisms that regulate the resistance to multiple ATR inhibitors" has been formally accepted for publication in PLOS Genetics! Your manuscript is now with our production department and you will be notified of the publication date in due course.

With kind regards,

Bailey Hanna

PLOS Genetics

On behalf of:
